# Analysis of the relationship between yield in cereals and remotely sensed fAPAR in the framework of monitoring drought impacts in Europe

Carmelo Cammalleri, Niall McCormick, Andrea Toreti

European Commission, Joint Research Centre (JRC), 21027 Ispra (VA), Italy.

*Correspondence to: Carmelo Cammalleri (carmelo.cammalleri@ec.europa.eu)*

**Abstract.** This study focuses on the relationship between satellite-measured fAPAR (Fraction of Absorbed Photosynthetically Active Radiation) and crop yield cereals in Europe. Different features of the relationship between annual yield and multiple time series of fAPAR, collected during different periods of the year, were investigated. The two key outcomes of the analysis are the identification of the period: i) from March to October as the one having the highest positive correlation between fAPAR and yield; ii) from February to May as the period characterized by most of the estimated negative correlation. While both periods align well with the commonly assumed dynamic of the growing season, spatial differences are also observed across Europe. On the one hand, the Mediterranean regions report the highest correlation values ($r > 0.8$) and the longest continuous periods with positive statistically significant results (up to 7 months), covering most of the growing season. On the other hand, the central European region is characterized by the most limited positive correlation values, with only 2 months or less showing statistically significant results. While marked differences on the overall capability to capture the full dynamic of yield are observed across Europe, fAPAR anomalies seem capable to discriminate low yield years from the rest in most of the cases.

**Keywords:** vegetation indices, agricultural drought, EDO, CEMS.

## 1. Introduction

Drought is a multifaceted phenomenon threatening societies, economies and ecosystems in a complex web of cascading effects (UNDRR, 2021). Amongst the major sectors that are impacted by drought, agriculture is still recognized as the most sensitive one (FAO, 2015; FAO et al., 2018; FAO, 2021), as reflected by the large share of reported impacts for agriculture over the majority of countries and drought events in Europe (Stahl et al., 2016).

Most drought monitoring systems recognize the prominent role of agricultural drought, by refining indicators of meteorological drought in order to better account for impacts on vegetation growth (e.g. the Standardized Precipitation-Evapotranspiration Index – or SPEI; Vicente-Serrano et al., 2012), and/or by directly incorporating drought indicators that are based on remotely sensed vegetation indices (WMO and GWP, 2016). In particular, negative deviations from climatological values of satellite measurements of vegetation "greenness" – for example, the standardized anomalies of the fraction of Absorbed Photosynthetically Active Radiation (fAPAR) that are provided by the European and Global Drought Observatories (EDO and GDO, https://edo.jrc.ec.europa.eu) – are often adopted as a proxy variable for the adverse effects of drought on vegetation.

While such approaches are logically based on the connection between reduced vegetation greenness and diminished plant productivity, it is also well known that droughts occurring during different phenological stages may have different impacts on yield and production (i.e. Barros et al., 2021; Ceglar et al., 2020; Chaves et al., 2002; Demirevska et al., 2009; Monteleone et al., 2022; Stallmann et al., 2020; Zampieri et al., 2017). Consequently, greenness anomalies are not always directly related to reduction in yield, depending on the development stages of the vegetative cycle when they manifest. Some studies have tried to account for this concept by limiting the analysis to the growing period and excluding data for the plant dormancy phase (e.g. Rojas et al., 2011), by deriving key variation metrics (i.e. amplitude, integral, maximum) from the full growing season (e.g. Kang et al., 2018), or by focusing only on key periods (i.e. a specific month) that have been shown to correlate well with deviations in annual yield for a given study area (Bachmair et al., 2018).

Within the framework of the near real-time monitoring of drought events, the task of evaluating and quantifying the actual relevance of an observed anomaly in vegetation greenness is complicated by the need to update continuously the status based on newly acquired data, without the benefit of the full picture of the complete vegetation cycle. This limits the possibility to implement some of the above-mentioned approaches as part of

operational drought monitoring systems, other than the simple masking of data acquired
outside of a pre-defined period (e.g. the growing season). An example of an early warning
system that accounts for the timing of the observed anomalies is the Anomaly Hot Spots of
Agricultural Production (ASAP) decision support system (Rembold et al., 2019), where the
seasonal progression (expansion, maturity, senescence) is explicitly considered in determining
the warning level.
As part of the shift in the drought risk management paradigm from a reactive to a
proactive approach, the move from simple hazard indicators to quantitative assessments of
risk and impacts is likely to be further integrated within modern early warning systems
(UNDRR, 2021). In this regard, independent estimates of actual drought impacts, such as the
information that can be derived from records of yield deviations for different crop types,
constitute a valuable reference. Unfortunately, these information are often collected at coarse
spatial resolution and they are available with a significant temporal delay. They are however
very valuable to assess if anomalies in vegetation indices can be used to detect the effects of
drought conditions, and how their robustness as proxy of yield reduction varies in space and
throughout the year. This can also enable the successive evaluation of the efficiency of
remotely sensed indicators as a proxy for the effect of drought on vegetated land, and the
refinement of their use as stress-forcing data for agro-economic models, for the assessment of
losses in agriculture due to droughts (García-León et al., 2021).
In this context, the primary goal of this study is to analyse to what extent the year-by-
year dynamics of yield in Europe can be explained by a regularly updated operational
vegetation drought indicator, in particular by the fAPAR anomalies produced by EDO. Yield
data for cereals, recorded by Eurostat, are here used as a starting quantity to produce records
of anomalies in yield at European scale. The spatio-temporal variations in the relationship
between dekadal (i.e. 10-day) fAPAR anomalies and yearly yield deviations can help in
identifying the periods of the year when fAPAR represents a reliable proxy information of
yield reduction impacts in Europe. This would prove a quantitative basis for improving the
assessment of drought impacts in agriculture, with potential benefits both for drought
monitoring systems and for agro-economic models.
**2. Material and Methods**
**2.1 Eurostat yield dataset**
Eurostat, the European Statistical Office, publishes regular reports of statistics on annual
crops, including data on production, cultivated area and yield for different crop types, at both
national and sub-national aggregation levels (Eurostat, 2020), with the aim of providing a
harmonized database of data collected by EU Member States and neighbouring countries.

For the purposes of this study, annual yield data of cereals (wheat and spelt, rye, barley,
oats, grain maize, triticale, and sorghum) have been retrieved between 2001 (first full year
with available fAPAR data) and 2018 (last available year in the Eurostat database at the time
of this study), mostly at the spatial scale of Eurostat's so-called "NUTS 2" regions (hereafter
referred to simply as regions). Only in the case of Germany and the UK, data at the NUTS 1
level were used, in order to maximize both data coverage and consistency in region size with
the rest of the domain.
Since yield data are to be used for computation of deviations from the long-term
average, temporal consistency in the data records is essential. For this reason, records that are
flagged by Eurostat as estimated, provisional, unreliable or with a definition that differs due to
missing components, were excluded from the analysis.
Systematic changes in the annual yield time series were removed by applying a
Savitzky–Golay filter to account for advancement in technology and crop management
(Tadesse et al., 2015), before standardized anomalies were computed only for those regions
with more than 9 years of data (i.e. half of the analyzed period). A linear de-trending was also
tested (not shown), but a limited effect of this choice was observed on the obtained yield
anomalies time series. Following this procedure, 240 regions with valid time series were
obtained (out of the 267 regions considered at the start of the study).
**2.2 MODIS fAPAR dataset**
The fraction of Absorbed Photosynthetically Active Radiation (fAPAR) is one of the 50
Essential Climate Variables recognized by the UN Global Climate Observing System
(GCOS), mainly thanks to its direct relationship with primary production
(https://gcos.wmo.int/en/essential-climate-variables/fapar).
fAPAR, and in particular its deviations from historical climatology, constitutes the ideal
proxy variable for the effects of drought on vegetated lands (Rossi et al., 2008). In this
context, remote sensing images collected by the MODIS (MODerate resolution Imaging
Spectroradiometer) sensor represent a unique data source for drought studies, due to the
unprecedented longevity of the Terra satellite.
In this study, the standard MODIS Terra LAI/fAPAR product (i.e. MOD15A2H,
Collection 6) is used (Myneni, 2015), in which global fAPAR maps are derived from the
atmospherically corrected Bidirectional Reflectance Distribution Function (BRDF) recorded
by MODIS in 7 spectral bands, by solving the three-dimensional radiation transfer process
through a look-up-table approach (Knyazikhin et al., 1998; Wang et al., 2001).

The standard MODIS product is distributed as 8-day composites (using a maximum

composite method) at a spatial resolution of 500-m in 1,200 × 1,200 km tiles on a sinusoidal
grid. Data include a quality assessment (QA) layer that allows to detect where the simplified
back-up algorithm has been used.

Datasets of both fAPAR and fAPAR anomalies based on MOD15A2H raw data are

regularly produced as part of the European and Global Drought Observatories (EDO and
GDO, https://edo.jrc.ec.europa.eu) of the EU's Copernicus Emergency Management Service.
The operational fAPAR dataset is obtained after a set of pre-processing procedures, including:
1) screening of the low-quality data based on the QA flag layer; 2) spatial aggregation of the
data (simple average) at 1-km resolution and re-projection onto a lat/lon regular grid at 0.01°
resolution with nearest neighbour resampling; 3) temporal aggregation at dekadal scale (three
maps per month: days 1–10, 11–20 and 21–end-of-month) by means of a weighted average of
the two closest 8-day images (weight proportional to the overlapping with the dekadal
period); and 4) exponential temporal smoothing of the dekadal data (with smoothing
parameter equal to 0.5; Brown and Meyer, 1961).

Here, the fAPAR anomalies were computed as standardized deviations from the

reference period (2001-2018), only if at least 6 years of data were available and only where
the long-term standard deviation was greater than 0.01 (to exclude areas of low variability,
such as deserts or highly stable densely vegetated areas). The reference period of 2001-2018
is consistent with the one used for yield anomalies.
**2.3 Analysis strategy**

In this study, the analysis of the relationship between the dekad time series of fAPAR

anomalies and yearly crop yield is based primarily on the Spearman correlation coefficient
(*r*). In order to carry out the analysis, the two main discrepancies between the two datasets,
namely regarding the spatial units (i.e. regions versus cells) and temporal frequency (year
versus dekad), must first be considered.

Given the focus of the study, the only fAPAR conditions that are relevant are the ones

observed over arable land. Therefore, the fAPAR anomaly data were first upscaled to NUTS 2
regions as a weighted average of all the 0.01° resolution fAPAR anomaly values within a
region, with a weighting factor based on the fraction of each grid-cell classified as arable land
according to the latest Corine land cover map (CLC2018, https://land.copernicus.eu/pan-
european/corine-land-cover/clc2018). This masking allows for removing from the NUTS 2
average all grid cells where the fAPAR dynamics are not related to agriculture (e.g. forest and
urban rural areas).

Regarding the temporal frequency, while fAPAR anomaly data are available throughout

the year, similar studies (e.g. Rojas et al., 2011) have focused only on data collected during
the growing season. A north-to-south gradient has been observed in the start, the end, and the
length of the growing season in Europe, with April-September being a common period all
over Europe, but with early start in February and late end in November over many areas
(Rötzer and Chmielewski, 2001). Estimations of the growing season directly based on
remotely sensed vegetation indices have also highlighted a very early start in the
Mediterranean, around October/November of the previous year (i.e. Atzberger et al., 2014),
likely related to combined effects (e.g. infesting weeds, early sowing and emergence) on the
remote sensing signal. Following these considerations, here we analyse an extended period,
testing the relationship between the yield of a particular year and the fAPAR anomalies
between the first dekad of October of the preceding year and the end of the current year, for a
total of 45 dekadal time series.

The set of correlation analyses between each of the 45 dekadal time series of fAPAR

anomalies and yearly yield data is used to construct a "correlogram", which relates the dekad
with the corresponding $r$ value. The example of Tuscany region in Italy (Fig. 1)  highlights
some common behaviours of the correlogram, such as a relative smooth transition between
periods of positive and negative $r$ values. Different analyses can be performed, depending on
the critical values that are extracted from these plots and on the goal of the analysis. Here, we
faced the problem in two different ways: a) detecting periods of similar behaviour and
accuracy but variable length; and b) detecting periods of similar length but variable accuracy
and behaviour.

For these two analyses, we distinguished between two different behaviours in the

fAPAR-yield relationship, a direct relationship (i.e. negative anomalies in fAPAR correspond
to negative anomalies in yield) and an inverse relationship. The latter may occur when a
strong vegetative growth is observed early in the season during drought years, especially in
energy-limited conditions (van Hateren et al., 2021). We also distinguished between two
levels of accuracy, statistically significant correlations ($p < 0.05$, either positive or negative)
and a less stringent condition where at least different than zero $r$ values (i.e. $|r| > 0.15$) are
considered. This second tier of values represent those conditions where a statistically
significant correlation (at $p < 0.05$) is not achieved, but a positive/negative relationship can
still be estimated. A value of 0.15 is used, as it corresponds to roughly 1/3 of $r$ at $p = 0.05$ in
the case of a full sample.
By defining a period as a streak of consecutive dekads of length ($L$) between 2 and 45,
990 periods of various length can be analyzed for each region, and for each of these periods
four main metrics (ranging between 0 and 1) are computed: 1) $F_{p+}$, the fraction of $r$ values in
the period that are positive and statistically significant (i.e. $r > 0$ and $p < 0.05$; 2) $F_{p-}$, the
fraction of $r$ values in the period that are negative and statistically significant (i.e. $r < 0$ and $p$
$< 0.05$); 3) $F_+$, the fraction of $r$ values in the period that are at least positive (i.e. $r > 0.15$), and
4) $F_-$, the fraction of $r$ values in the period that are at least negative (i.e. $r < -0.15$). By
definition, $F_+$ and $F_-$ are always greater equal than $F_{p+}$ and $F_{p-}$, respectively. We can then
focus on the longest periods (among the 990 periods) having homogeneous behaviour and
accuracy for a given region (homogeneous periods hereafter), e.g. a period with $F_{p+} = 1$. Due
to the smooth dynamics observed in most correlograms, these homogeneous periods are rather
well defined. In the rare instances when multiple homogeneous periods of the same length are
found for a region, the period closer to the surrounding regions is selected.
In the example reported in Figure 1, the dekads between 23 and 36 (light grey area) are
clearly part of the longest period with all positive and statistically significant $r$ values, $L = 14$,
while the dark grey area demarks the longest period with $F_- = 1$ ($L = 6$).
A further set of analyses is focused instead on a fixed time window selected among a
limited range of lengths of the periods (i.e. a subset of periods among the 990 possible periods
with length from 2 to 45). The boundary values of this subset of periods can be derived from
the previous tests. Within these limits, an optimal positive (negative) period for each region
can be defined as the period with the maximum (minimum) average $r$ value. Differently from
the first group of analyses, these optimal periods have varying $F_{p+}$ and $F_+$ ($F_{p+}$ and $F_+$) values
(corresponding to the average $r$ value) that can be used to quantify the robustness of the
relationship between fAPAR and yield. This analysis is performed on a subset of periods to
avoid selecting as optimal very short periods (i.e. of length 2) for regions with a prominent
peak value, or very long periods for regions where the correlogram is particularly flat.
Finally, while the analyses based on correlation give an insight on the relationship
between fAPAR and yield over the full spectrum of variability, a further test focused only on
extreme low yields is also performed, given that in the context of drought monitoring it would
be sufficient to be able to distinguish these conditions from the rest in order to successively
detect the drought-affected years. Here, the total number of cells for each region with fAPAR
anomalies $< -1$ (a common threshold used in extreme analyses) is computed during low yield
years (yield anomalies < -1), and it is compared with the same during the other years (yield
anomalies ≥ -1). The assumption of this analysis is that the ratio of these two quantities should
be greater than one in the case of a direct relationship.
**3. Results**
**3.1 Dynamics of yield anomalies and relationship with droughts**
While negative anomalies in yield can be often associated to drought events, the full dynamic
of standardized yield anomalies for cereals, as described in Section 2.1, cannot be exclusively
ascribed to the occurrence of drought conditions. However, the ability to capture the year-by-
year dynamics of yield using fAPAR anomalies is here evaluated with the goal of exploiting
this relationship in the framework of drought monitoring, hence the connection between low
yields and droughts need to be firstly assessed.
Figure 2 depicts the temporal evolution of yearly yield deviations, highlighting some
clear spatial patterns of significantly negative anomalies (i.e. yield anomaly < -1). Following a
review of the scientific literature for past drought events, it is possible to associate a
documented main drought event to many of these large clusters, as summarized in Table 1.
Seven main droughts are reported, ranging from the well-known drought in central Europe of
2003 (Rebetez et al., 2006) to the central-north European drought of 2018 (Buras et al., 2020;
Toreti et al., 2019).
The existence of a cause-effect relationship between these largest spatial patterns
observed in negative yield anomalies and the listed major drought events is further supported
by the study of Spinoni et al. (2015), which categorized the listed events (except the last two,
which occurred after that study) as being among the most severe in Europe according to
meteorological drought indices.
For each of the drought events listed in Table 1, specific independent scientific
references are also provided, which include details on the evolution of the meteorological
conditions, and the potential impacts on agriculture. Overall, analyses of these data tend to
support that the adopted dataset of yield anomalies shows the impacts on vegetation of the
major European droughts, in conformance with the conclusions of other studies at regional
level in Europe (Bachmair et al., 2018; Potopová et al., 2015), or for other parts of the world
(e.g. Yang et al., 2020).
**3.2 Detection of the homogeneous periods in the fAPAR-yield relationship**
While many studies focused on the local maximum $r$ value to detect when and where
fAPAR and annual yield anomalies best correlate, isolated peak values may alter the
perception of the robustness of fAPAR as a proxy variable of yield. In the context of an
operational drought monitoring system, where continuous estimates should be provided rather
than "one shot" predictions, information on longer homogeneous time periods are more
valuable.
Focusing first on the positive $r$ values, we analysed the periods with only statistically
significant values ($F_{p+} = 1$), or only at least positive values ($F_+ = 1$). The maps in Figure 3
reports the local maximum lengths corresponding to these two quantities, namely positive
homogeneous periods. Both of these maps show generally longer homogeneous periods in
southern Europe, with the largest values observed for some Mediterranean regions (e.g. most
of Spain, Cyprus, Sicily, Apulia and the Aegean/Mediterranean Turkey), and the smallest
values (or no homogeneous period at all) mostly located in Central Europe (i.e. Germany,
Poland and north-eastern France). On average, the maximum length of the periods with $F_{p+}$
=1 is limited in most of the cases (5.5 ± 4.3 dek, almost 2 months), whereas the values more
than double in the case of $F_+ = 1$ (13.0 ± 8.3 dek, more than 4 months).
Generally, almost all the maximum $r$ values in the correlograms are obtained in the
dekads between mid-February and mid-September, which is expected since this period aligns
well with what is commonly considered the growing season in Europe (Atzberger et al., 2014;
Rötzer and Chmielewski, 2001). Nonetheless, a large variability in the length of both positive
homogeneous periods is observed, with southern and central Europe confirmed to be not only
the areas with highest and lowest $r$ values, respectively, but also the areas with the longest (i.e.
4-7 months) and shortest (up to 2 months) periods with consecutive statistically significant
positive correlations.
Due to the large variability in the length of the homogeneous periods observed in Figure
3, a direct analysis of the spatial patterns in the starting and ending dekads is not feasible. So,
in order to evaluate synthetically the temporal location of these homogeneous periods, we
analyzed which dekads each of them covers, and computed for every dekad the fraction of
NUTS 2 regions (out of 240) that includes that particular dekad in the homogeneous period
(Fig. 4). For example, dekad 27 (i.e. the first dekad of July starting from the beginning of
October of the previous year) is part of the maximum homogeneous period in about 20% and
50% of the regions, for $F_{p+}$ and $F_+$, respectively. It is worth noting that about 21% of the
NUTS 2 regions do not have a period (minimum 2 consecutive dekads) with $F_{p+} = 1$.
It is possible to observe two "flexing points" in each of the two time series in Figure 4
around 0.1 for $F_{p+}$ and 0.2 for $F_+$. Starting from these values, we can detect two optimal
homogeneous periods: from end-of-April to mid-October (6 months) for $F_{p+}$, and from March
to early-November (8 months) for $F_+$.
Moving to the negative correlation values, two maps analogous to the ones in Figure 3
are reported in Figure 5 for $F_{p-}$ (panel a) and $F_-$ (panel b). These two maps show how the
longest negative homogeneous periods are in general shorter than the ones for positive
correlations, with an average value of $3.0 \pm 1.6$ dekads for $F_{p-}$ and $7.0 \pm 3.9$ for $F_-$. The lack of
statistically significant negative $r$ values is especially evident, with almost 50% of the regions
having no homogeneous periods with $F_{p-} = 1$. The map for $F_-$ (Fig. 5b) allows for some
additional considerations on the spatial distribution, with moderate maximum lengths (around
9 dekads) in most of western and central Europe, and some high values (higher than 15
dekads) in some regions of southern Europe.
In terms of temporal distribution, the histograms on Figure 6 depict the fraction of
NUTS 2 regions that includes that particular dekad in the negative homogeneous periods.
Overall, the fraction values are lower than the ones observed for the positive periods (see Fig.
4), with two distinguishable peak periods in the $F_-$ values, the first in early season (February-
May) and the second after the end of the season (October-December, sowing period for the
winter crops).
Most of the homogeneous periods early in the season correspond to regions in western
and southern Europe, and the late season periods are mostly located in central and northern
Europe. In the framework of drought monitoring, the first can be potentially exploited as early
warning signals of subsequent reduction in fAPAR due to drought (as seen in the positive
homogenous periods that usually follows in the correlograms). The second mostly occur right
after the harvesting season, and hence as no value for early warning systems.
**3.3 Performance for a fixed time-window**
A clear outcome of the previous analyses is that the length of the homogeneous periods
with negative correlations is limited compared to the positive correlations, and mostly useful
for drought monitoring only early in the growing season. Therefore, we focus only on the
positive correlation values for the successive analyses. The two lengths (6 and 8 months)
derived from the data depicted in Fig. 4 are used as the minimum and maximum boundary
values to find the local optimal period for each region (see Section 2.3).
The results of this bounded analysis of the local optimal period are shown in Fig. 7,
where the starting dekad ($d_i$, panel a) and ending dekad ($d_e$, panel b) of the optimal period are
depicted for every region. Fig. 7a shows a general pattern of an early start in Central Europe
(i.e. February/March), and in few southern regions of the Mediterranean, and a late start (i.e.
May/June) in most of southern and western Europe. This late start is of course in line with the
previously observed negative correlations in February/May over the same regions.
Analogously, Fig. 7b shows that the end of the optimal period occurs mostly around
October/November, after the harvesting, in both southern and western Europe, and
August/September in central Europe, with then mostly negative correlations in central and
north Europe occurring after this period (likely due to spurious correlations).
Given that these optimal periods have been derived based on the average $r$ values in the
6 to 8-month period, the $F_{p+}$ and $F_+$ values corresponding to these optimal periods can assume
any values between 0 and 1 (no significant/positive $r$ values to all significant/positive $r$ values
within the optimal periods). For this reason, we classified each region based on the combined
values of these two metrics, as represented by the legend included in Fig. 8. In this map, the
green areas show a good capability to reproduce the dynamic of yield deviation for the whole
optimal period (the fraction of high $r$ values in the two optimal periods is high), with the
regions in dark green having the overall best performance (over half of dekads with
statistically significant $r$ values and more than 2/3 with at least positive values). Conversely,
the red regions show a poor capability of the fAPAR anomalies to capture the yield dynamics,
with the dark red regions having less than 1/10 of statistically significant values (i.e. less than
a month) and less than 1/3 of positive correlations during the optimal period.
Overall, slightly more than half (i.e. 55.8%) of the study regions are classified in one of
the green classes, with a predominance of these regions in Mediterranean and south-eastern
Europe. The rest of the study area is almost equally split between regions with average
performance (yellow class, 23.3%), and poor performance (red classes, 20.9%). Among the
red classes, the majority of the regions fall in the category with intermediate $F_+$ values (1/3 <
$F_+$ < 2/3) but low statistically significance ($F_{p+}$ < 1/10). Most of these regions are located in
central Europe, between northern France, the United Kingdom, Germany and Poland.
Spain stands out as having particularly robust performances, even among the generally
good performing Mediterranean area. While the start and end of the optimal period varies
across the area (March to May, and September to November, respectively), the results are
consistently in the best class (dark green in Fig. 8). Among the Mediterranean countries, some
mixed results can be observed in Italy and Greece.

**3.4 Detection of low yield years**

The previous analyses show a noticeable difference in the performance of fAPAR anomalies to capture the full range of variability of yield anomalies across Europe, as quantified by the results on the optimal periods summarize in Figure 8. For the same optimal periods, the number of fAPAR anomalies < -1 were cumulated for low yield years (yield anomaly < -1) and the other years, separately, and the ratio between these two quantities is depicted in Figure 9.

Overall, values greater than 1 are observed over most of Europe in Figure 9, suggesting a good performance of fAPAR anomalies to detect extreme low conditions in annual yield. While the ratio is only slightly higher than one in some regions where the previous analyses highlight poor performances (i.e. the UK and France), years with severe reductions in yield are still well-captured by fAPAR.

Finally, the plot in Figure 10 shows a comparison between the ratio computed on the optimal period (grey area) and the one computed on the full year (all 36 dekads, black area). Since the years are divided in the two categories based on yield data, the size of the two datasets is independent from the selected period (optimal or full year), making the intercomparison straightforward. The plot show an overall increase in the ratio when only the dekads in the optimal period are considered, which translate in a better ability to discriminate low yield years compared to simply account for all the anomalies observed across the full year.

**4. Discussion**

The value of the results reported in the previous section in the context of drought monitoring is related to the assumption that anomalies of cereal yields show the effects of drought on vegetation during drought years, as demonstrated for example by Brás et al. (2021) who quantified an approximately 9% reduction in European cereal yields due to historical droughts (1961-2018), with an increasing intensity in more recent years. The spatial patterns in negative yield anomalies for the dataset used in this study, and the cross comparison with documented past drought events, confirm the general assumption that low yields are recorded during drought years, even if not all the low yield values may be associated to droughts. These data confirm that understanding the role of fAPAR as proxy of yield is valuable for drought monitoring, even if a non exclusive correspondence between low yield/fAPAR and drought exists.

Due to the focus on data commonly used in operational drought monitoring systems, a
common element for all the performed analyses is the independent use of each dekadal
fAPAR time series. While different results may be achieved by using metrics based on the full
growing season (e.g. Kang et al., 2018), such analyses are not easily transferable to a near-real
time monitoring framework. Overall, the correlation coefficients computed using fAPAR
collected during multiple dekads suggests a predominance of positive values over all regions.
This is in line with the expected direct relationship between fAPAR and yield during the core
growing season, as well as with most of the past studies which focused primarily on the
positive correlation. Indeed, most of the maximum values of correlation seems to be located
within the conventional growing season, and the south-north gradient observed in both of the
positive homogeneous period maps (Fig. 3) is in broad agreement with the expected
increasing gradient in growing season length observed over Europe (Rötzer and Chmielewski,
2001). However, there is not a perfect matching between the growing seasons and the periods
with higher correlation values, and while studies on satellite-derived phenology have detected
growing season lengths ranging from 5 to 9 months (Rötzer and Chmielewski, 2001), the
average length of the periods with positive and statistically significant correlations seems to
be shorter.

Consistently high positive correlation values are obtained over most of Spain, in line
with a recent study over the region (García-León et al., 2019), which reported good
performances of the satellite-based Vegetation Condition Index (VCI) for different type of
cereals, especially for winter wheat and barley. Over central Italy, Todisco et al. (2008)
observed good correlation between yield in sunflower and sorghum with common drought
indices (Standardized Precipitation Index, SPI, and Soil Moisture Severity Index), with a
maximum correlation around weeks 27-29 of the growing season (i.e. July) and statistically
significant values for periods ranging from 2 to 4 months. Similar timing, but with a slightly
shorter optimal length, has been observed in our analysis for the same area.

For Germany, Bachmair et al. (2018) found significant correlation values between VCI
and Vegetation Health Index (VHI) anomalies in the month of August, and yield deviations
for maize, that are comparable with the maximum values observed for western Germany in
our study. A mix of high correlation and missing data is reported in that study for eastern
Germany, where our results are statistically significant only for a very limited period. These
differences may be explained by the focus on specific crop types (not included in our study),
as the same authors also highlight how the accuracy of their relationships varied for the
different crops.

Similar to our results, Labudová et al. (2017) found significant correlation with SPI and
Standardized Precipitation Evapotranspiration Index (SPEI) in the Danubian lowlands only
for summer months, or for a very limited time (i.e. June) in the Eastern Slovak lowlands. For
these regions, the values of the maximum homogeneous period with $F_{p+}$ =1 ranged between 3
and 9 dekads as shown in Fig. 3.

The good results observed over the western Mediterranean and the countries around the
Black Sea are in agreement with the founding of López-Lozano et al. (2015), which reported a
similar pattern in their study based on a different fAPAR product (derived from SPOT-VGT
data). This seems to suggest that the observed relationship are likely independent from the
data source, and more intrinsically connected to the capability of the physical quantity
fAPAR to reflect the variation in yield under certain conditions.

The presence of limited periods with consecutive negative correlations early in the
growing season may be related to the lagged response of vegetation to water deficits (Crow et
al., 2012), which results in positive greenness anomalies early in the season followed by
negative values later on (i.e. delay in the phenological cycle). Another explanation can be the
limited immediate effect of water deficit during energy-limited periods (Zscheischler et al.,
2015), which can also be the reason behind the general poor correlation between fAPAR and
yield over regions where water is not a key limiting factor. This inverse relationship observed
early in the season is currently under-explored in drought monitoring systems, which mostly
focus on the direct relationship, and it may have an interesting role as an early warning tool
under specific conditions. However, the results obtained in this study suggest a limited
temporal extension and statistical robustness of the periods with inverse relationships, which
usually are followed by much longer and robust periods of direct relationship.

The late start of the optimal period in many regions of the Mediterranean and western
Europe, compared to the rest of the domain, is associate to the presence of these periods of
inverse relationship early in the growing season. Given the particular climate of the
Mediterranean region, and the key role of dry and hot spring-summer months in propagating
the water deficits in the area, a lagged response in vegetation is expected. In contrast, Central
Europe is characterized by an earlier start of the optimal period (March to August), compared
to the Mediterranean and western Europe, that seems to precede the expected growing season
(June to October), further stressing the imperfect match between optimal period and growing
season. For central Europe, Potopová et al. (2015) found high yield-drought correlation for
cereals (better than other crops) over Czech Republic between April-June, a result in line with
our findings. The late start (April/May) in the northern regions of Scandinavia compared to
central Europe, is mostly explained by the lack of reliable fAPAR data earlier in the year, due
to low sun angles.

Focusing on the optimal period, mixed performances are obtained in Italy, with low

agreement particularly in Sardinia and regions along the Apennine mountains. Although
García-León et al. (2021) found a positive relationship between annual-cumulated fAPAR
anomalies and yield for most main crop types, the aggregation of the results at national scale
does not allow the detection of differences among regions. Given the complex morphology of
those regions, potential unreliability in the fAPAR estimates may be a possible cause for the
poor performances. Complex morphology can also be the reason for poor results over few
other Mediterranean areas, such as Greece.

The spatial variability of the dependence of yield to water-limiting factors can be one of

the explanation of the observed patterns, with stronger correlation between fAPAR and yield
over water-limited regions (Zampieri et al. 2017), and weaker relationships over regions
where other factors may play a major role in controlling yield rather than simply greenness
dynamics. Similar considerations were also made by López-Lozano et al. (2015), even when
results are disentangle between different crop types (wheat, barley and maize).

Indeed, another possible contributing factor underlying the spatial differences in the

retrieved optimal periods can be the potentially variable response of different cereal types
included within the overall cereals Eurostat category. Since different predominant cereal types
are cultivated locally, this variability can also contribute to the observed spatial variability in
the results. This is supported by other studies that have demonstrated different responses for
different crop types (García-León et al., 2021; Labudová et al., 2017). While applying the
analysis to different cereals sub-categories, or even different plant types, may be useful to
understand better the relationship between fAPAR and yield for each specific crop, the results
of this study for all cereals provide valuable experimental information on optimal periods that
can be more easily integrated into an operational drought monitoring system, which does not
only focus on agricultural drought impacts.


**5. Summary and Conclusions**
In this study, records of annual crop yield data for cereals were used to evaluate the
performance of satellite-derived fAPAR time series data in capturing year-by-year variations
in crop production for different periods of the year and growth stages of vegetation, given that
fAPAR anomalies (or other greenness indices) are often used in drought studies to capture the
effect of drought events on vegetation in absence of yield data.
Overall, the analysis of the correlograms computed by plotting anomalies of dekadal
fAPAR values against yearly yield deviations, was used for three main purposes:
▪ Investigation on continuous streaks of dekads with homogeneous behaviour (direct *vs.*
inverse) and agreement (i.e. statistical significance) but with different temporal length.
▪ Investigation of fixed length (6 to 8 months) optimal periods, defined as function of
the maximum average *r* within the given range of lengths.
▪ Evaluation of the capability of fAPAR anomalies during the optimal periods to
discriminate between low yield and other years.
The analyses confirm the period March to October as being the most relevant to
positively correlate anomalies of fAPAR and crop yield, being the period when most of the
highest values of correlation are estimated, and when most of the continuous periods with
statistically significant and positive *r* values are located. There is a generally good agreement
between these findings an both the duration and temporal location of the commonly defined
growing seasons in Europe, even if spatial patterns in periods with positive correlations and
growing season can also be rather different. While some periods with consistent negative
correlations are also observed between February and May, these are generally limited in
length to be considered as primary source of information to reproduce yield dynamics, but
they have potential as valuable early warning information.
The average growing period in Europe is usually characterized by a marked south-to-
north gradient, which is also observed in our analysis of the 6- to 8-month optimal periods
based on average *r* values. Some clear spatial patterns emerge in this analysis, such as the
early start in most of central Europe and the southern Mediterranean, and the late start in
southern and western Europe. These spatial patterns do not exactly match commonly observed
satellite-derived growing seasons, so they provide an independent assessment of which phases
of the phenological cycle are more valuable to capture yield variations, a valuable information
that can be incorporated into operational drought monitoring systems.
Another key output of the study is the generally good correlation between fAPAR
anomalies and crop yield anomalies over most of the Mediterranean regions and across the
full range of variability of yield data. This result can be explained by the strong dependency
of both yield and vegetation greenness to water-limiting factors, as also suggested by López-
Lozano et al. (2015) and Zampieri et al. (2017). Given the well documented high vulnerability
of this region to drought and the increasing threat posed by climate change (Cammalleri et al.,
2020; Dubrovský et al., 2014), this result suggests the possibility to link satellite-observed
fAPAR anomalies with actual impacts in agriculture, as a promising new development that
merits further exploration.
This study also highlighted the overall limited correlation, outside of very short time
periods, between fAPAR and yield over most of the NUTS 2 regions in central Europe.
Further analyses may be needed to better understand the reason behind this result. In this
context, a recent study by Beillouin et al. (2020) has demonstrated how simple climate
variables (i.e. high temperature and low precipitation) can explain much of the yield
variability in central Europe, in contrast with the situation in southern Europe. It is important
to further remark that even over these regions where the overall performance is limited,
fAPAR anomalies are still successful in discriminating between low yield years and the rest,
which is still a relevant feature to be further exploited in drought monitoring systems.

**Data availability:** The fAPAR dataset used in this study can be retrieved from the JRC Data
catalogue (EDO, 2021).

**Author contribution:** CC designed the experiments with inputs from AT and NMC. CC
developed the codes and performed the analyses. CC prepared the manuscript with
contributions and revisions from all co-authors.

**Competing interests:** The authors declare that they have no conflict of interest.

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

## Tables

**Table 1.** Main European drought events between 2001 and 2018 corresponding to the large patterns in negative yield anomalies (< -1) observed in the maps reported in Fig. 2. References to the scientific literature of each event are also report, with a brief description of the document impacts for the agriculture sector.

| Year of drought event | Area affected | Impacts for the agriculture sector | Reference |
|---|---|---|---|
| 2003 | Central Europe | Fall in EU cereal production of more than 23 million tonnes as compared to 2002. Also fodder deficit ranging between 30% and 60%. | Rebetez et al. (2006); De Bono et al., 2004. |
| 2005 | Iberia Peninsula | Cereal production reduced to 60% of average and severe shortage of wheat (more than 50% in Portugal). | García-Herrera et al. (2007); Gouveia et al. (2009). |
| 2006 | North-Eastern Europe | Crop yield losses and forest fires in Lithuania. About 20% yield reduction for all cereals in Poland. | Valiukas (2015); Somorowska (2016); Sassenrath et al. (2012). |
| 2007 | Eastern Europe | The drought destroyed 60% of the cereal crops in Romania, and lowest recorded | Bogdan et al. (2008); Sima et al. (2015); Demuth (2009). |

| | | yields in some counties. Estimated economic costs of at least 1.5 billion Euros. | |
|---|---|---|---|
| 2012 | Eastern Europe | About 5.9 million hectares of crops impacted all over Romania. | Sima et al. (2015) |
| 2017 | Southern Europe | Reduction in agricultural production, especially for cereals (among other crops) in Spain and Italy, with estimated losses of 2 billion euros in Italy. | García-Herrera et al. (2019) |
| 2018 | Central-Northern Europe | Yield reductions from 9% to 50% for the main crops. | Buras et al. (2020); Toreti et al. (2019) |

750

**Figures**

752

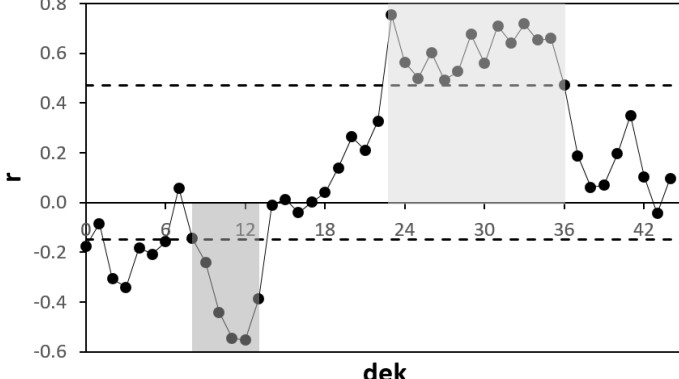

753

**Fig. 1.** Example of correlogram for one NUTS 2 region in Italy (ITI1, Tuscany). Each value represents the Spearman correlation coefficient between the fAPAR anomaly time series of a specific dekad and the yearly yield anomalies. The two horizontal dashed lines represent, respectively, the threshold for positive statistical significant value at $p = 0.05$ and the minimum negative threshold ($r = -0.15$, see the main text). Dekads are defined starting from the first one of October of the previous year (e.g. dek = 23 refers to the last dekad of May of the current year).

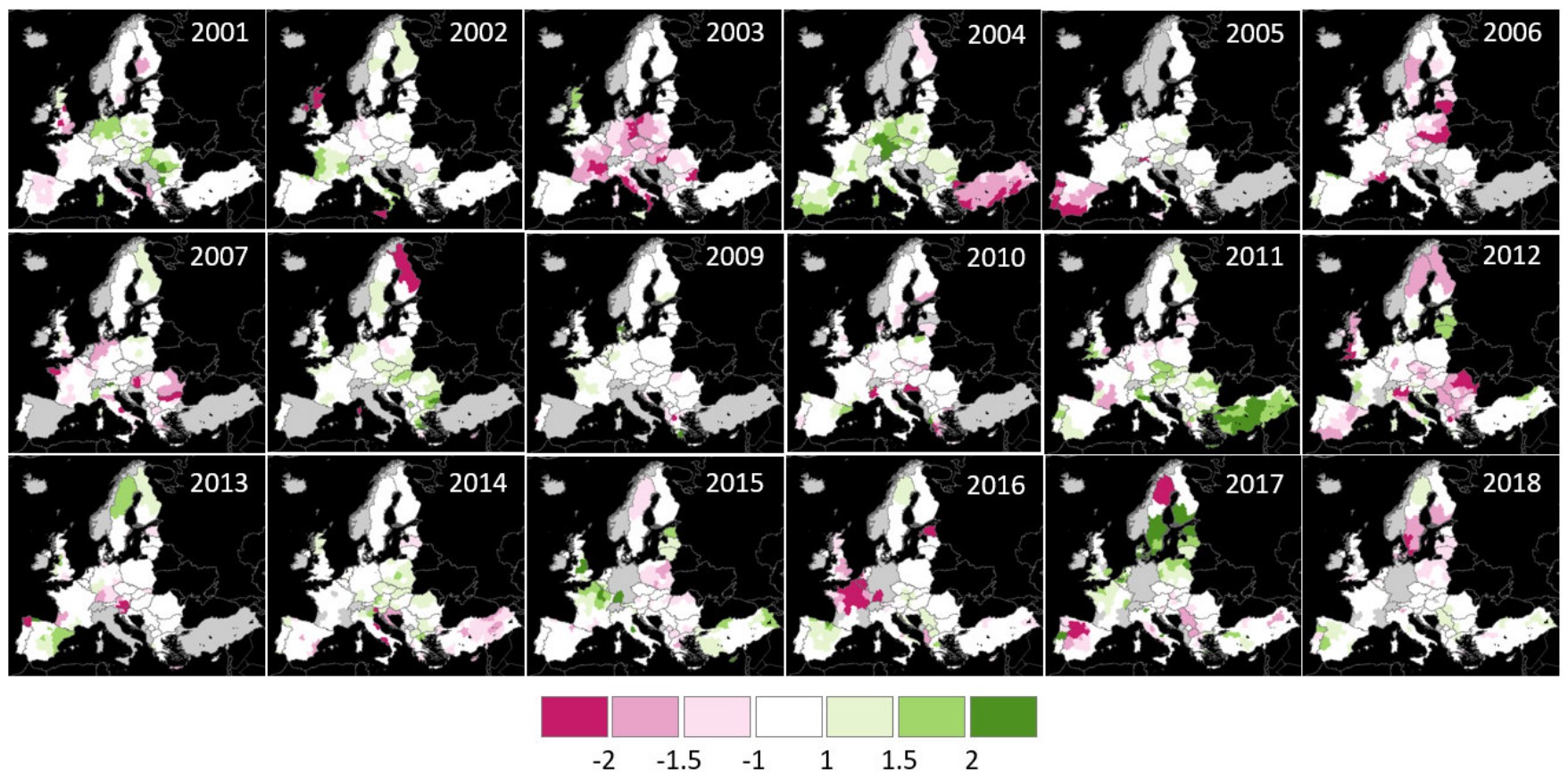


**Fig. 2.** Spatial distribution of annual standardized yield anomalies for the period 2001-2018. Anomalies are mapped at NUTS 2 level, with the exception of the areas detailed in Section 2.1. Data in grey are missing.

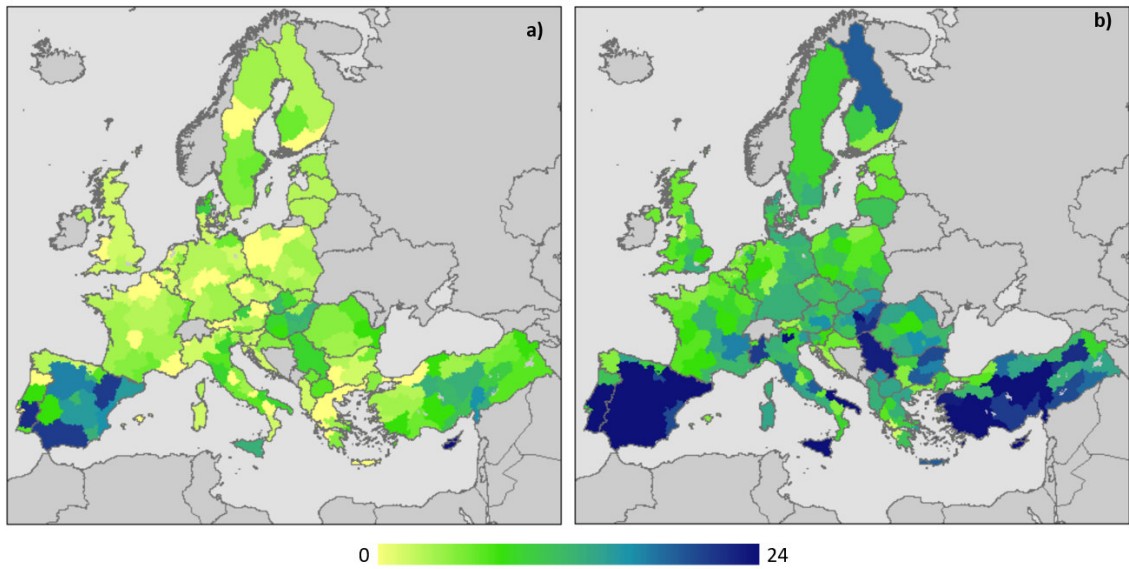


**Fig. 3.** Spatial distribution of the length (in dekads) of the longest period with $F_{p+} = 1$ (panel
a) and $F_+ = 1$ (panel b).

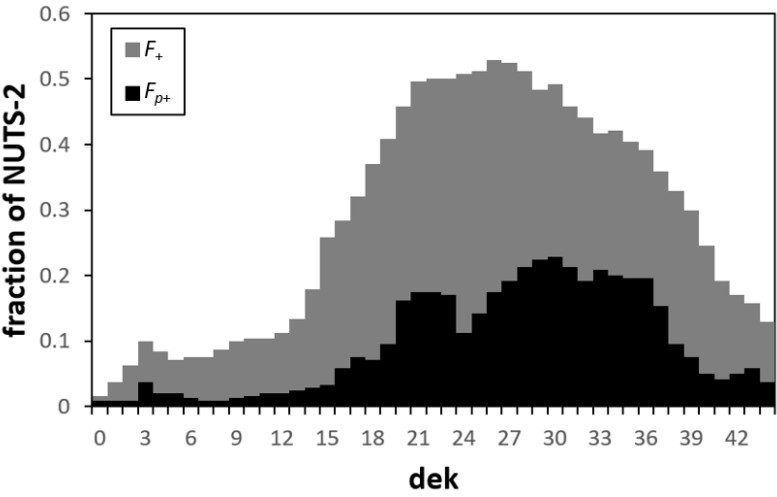


**Fig. 4.** Fraction of NUTS 2 regions for which each dekad is included in the longest
homogeneous period with $F_{p+} = 1$ (black) or $F_+ = 1$ (grey).

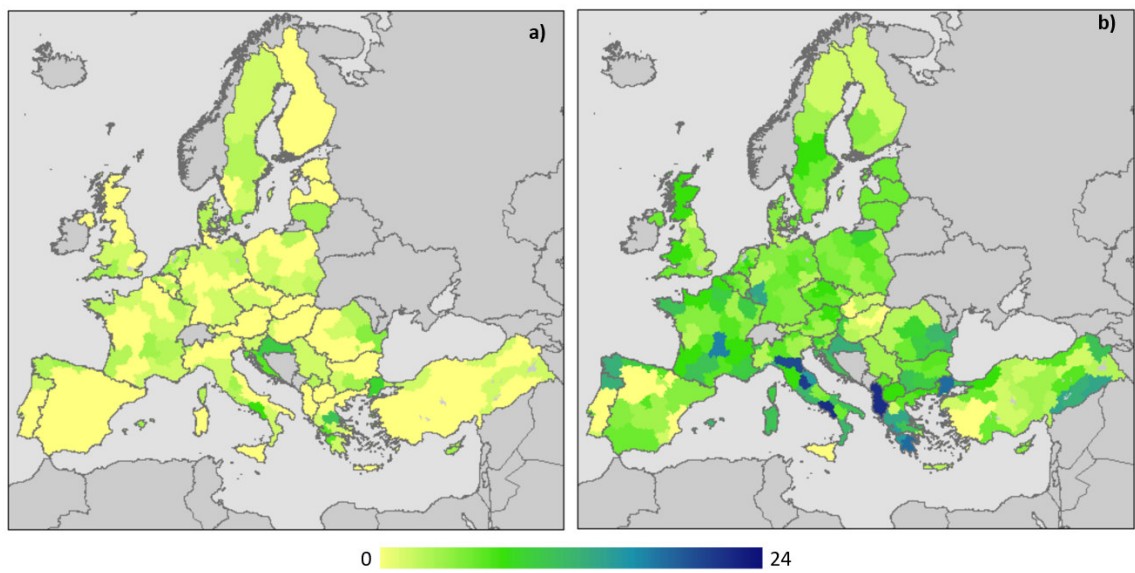


**Fig. 5.** Spatial distribution of the length (in dekads) of the longest period with $F_{p-} = 1$ (panel
a) and $F_{-} = 1$ (panel b).

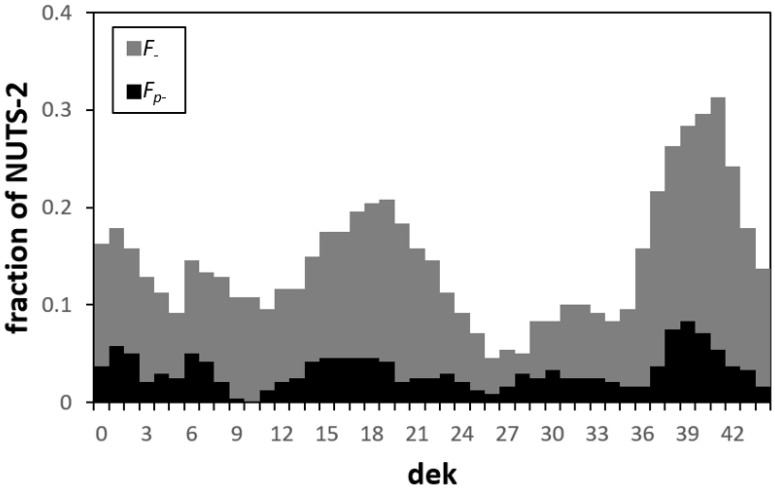


**Fig. 6.** Fraction of NUTS 2 regions for which each dekad is included in the longest
homogeneous period with $F_{p-} = 1$ (black) or $F_- = 1$ (grey).

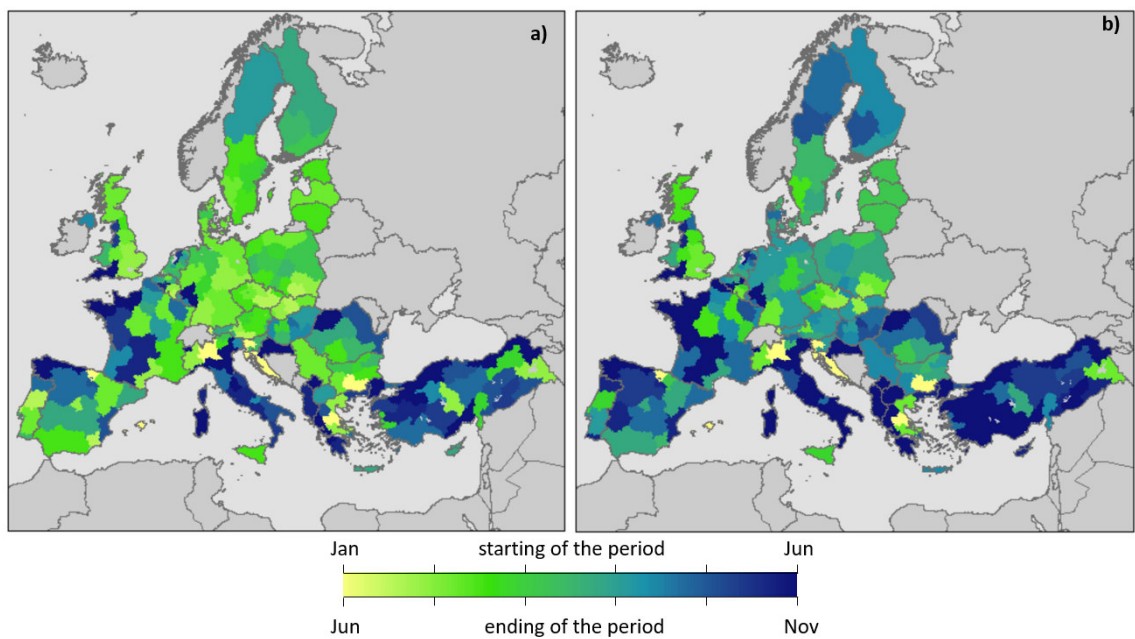

**Fig. 7.** Spatial distribution of (a) the starting dekad, and (b) the ending dekad, of the local optimal period based on the average correlation and bounded by a length from 6 to 8 months.

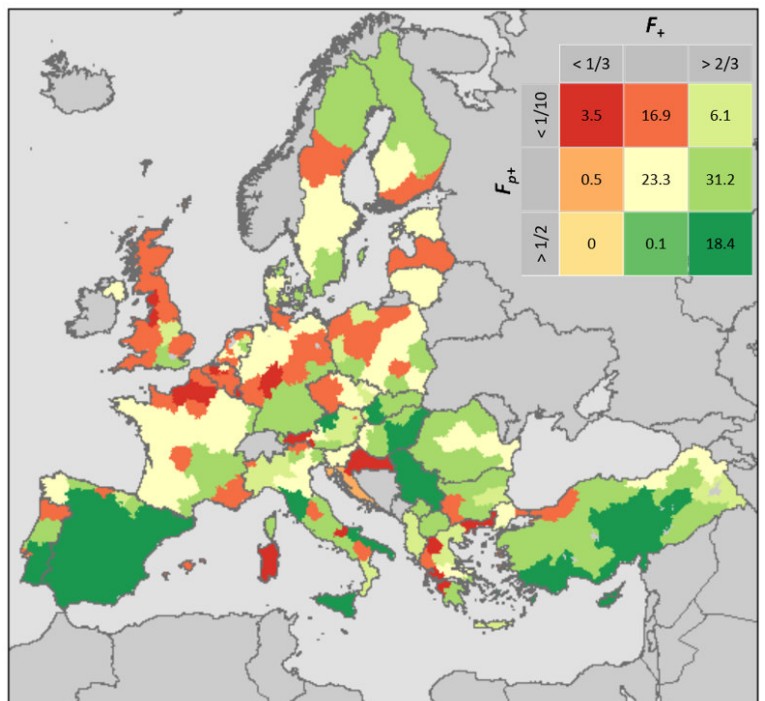

Fig. 8. Synthetic representation of the performance of dekadal fAPAR anomalies in reproducing the yearly yield variations during the local optimal period. The inserted legend shows the values of $F_{p+}$ and $F_+$ for each category, with the numbers inside each square representing the percentage (%) of the total NUTS 2 regions (out of 240) that falls under each category.

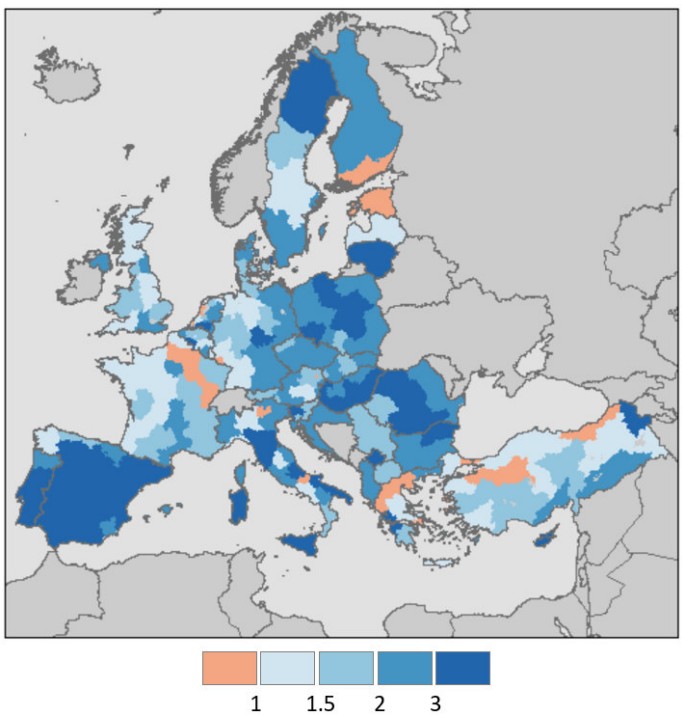


**Fig. 9.** Spatial distribution of the ratio between the number of fAPAR anomalies < -1 in the
optimal period (see section 3.3) during low yield years (yield anomalies < -1) and other years
(yield anomalies ≥ -1).

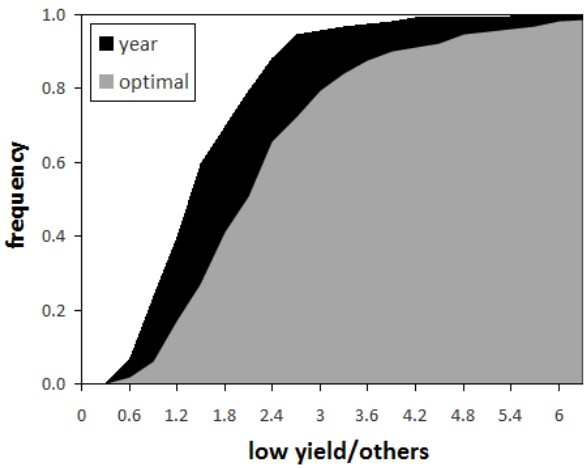


**Fig. 10.** Cumulated frequency of: i) the ratio between the number of fAPAR anomalies < -1 in
the optimal period (see section 3.3) during low yield years (yield anomalies < -1) and other
years (yield anomalies ≥ -1) (optimal, grey area); and ii) the ratio between the number of
fAPAR anomalies < -1 in the full year (36 dekads) during low yield years and other years
(year, black area).