# Peer review of "Analysis of the relationship between yield in cereals and remotely sensed fAPAR in the framework of monitoring drought impacts in Europe"

_Natural Hazards and Earth System Sciences, 2022_

## Author Response (AR1)

**Reviewer #1**

**General comments:**

The paper evaluates the relationship between cereal yields and the Fraction of Absorbed Photosynthetically Active Radiation (fAPAR, a satellite derived product) in Europe with the aim of capturing the effects of droughts on crop production. The anomalies of fAPAR were plotted against the yearly yield deviations. The correlation between fAPAR and yield anomalies is positive between the period from March to October, which corresponds to the cereals growing season in Europe. Some negative correlations were observed between February and May; they are limited in length but their analysis could be interesting to assess if they can be considered valuable early warning information. The average growing period in Europe is characterized by a marked south to north gradient; the season has an early start in central Europe and in southern Mediterranean, while it has a late start in southern and Western Europe. An interesting result of the study is the good correlation between fAPAR anomalies and crop yield anomalies over the Mediterranean, while the correlation is limited in most of regions of Central Europe. fAPAR anomalies are found to be useful to distinguish between drought and no/drought years in the majority of situations in which yield anomalies are used as proxies for drought impacts on agriculture.

Overall, the paper represents a good contribution to the understanding of drought and its effects on crop yield. The method applied is valid and the results are appropriately discussed. The work is well structured, the figures quality is good and the number of figures and tables is appropriate.

We want to thanks the reviewer for thoroughly analyze our research and for appreciating our manuscript.

**Specific comments:**

Line 32-33: Together with the two FAO reports of the 2015 and 2018 I suggest citing a most recent one: "The impact of disasters and crises on agriculture and food security: 2021" (FAO, 2021).

We added this reference to the revised version of the manuscript.

Line 48: I suggest citing a recent study (Monteleone et al., 2022) evaluating the effect of drought on different phenological stages of maize in Italy.

Thanks for the suggestion. We added this recent study to the references.

Line 94: Could you specify which crops Eurostat includes in the definition of "cereals"? Do you believe that the same results discussed in your study could have been obtained considering crops different from cereals?

We added some details on the definition of cereals adopted by Eurostat.

In the Conclusions, we highlighted how different results can be obtained by focusing on specific crops (even within the cereals category), as highlighted in some cases by local studies. However, such analysis goes beyond the scope of the study, which aims at giving operational insights to be integrated in a drought monitoring system not specifically focused on agriculture. We further stressed this concept in the revised version of the manuscript.

Discussion: I suggest comparing the obtained results with the ones reported in (López-Lozano et al., 2015), who found a significant correlation between fAPAR and official yields (R2>0.6) in water-limited yield agro-climatic conditions (e.g. the Black Sea region and the Mediterranean basin) for wheat, barley and grain maize.

We added a paragraph comparing our results with the ones from our colleagues, highlighting the similarities in the general outcomes but also the differences due to the different focus.

Table 1: It could be useful to add a column where the drought impacts on agriculture described in the cited studies are briefly summarized.

We agree with your suggestion, and we added some details on the impacts of drought on agriculture for those events.

Fig. 2: NUTS2 regions are quite difficult to visualize; I suggest to increase the size of the various panels and eventually having a sort of table with three columns and six rows instead of the actual figure in which there are six columns and three rows.

We decided to do not report the border of the NUTS2 regions to avoid an over-saturation of the figure. We understand your complain, and in order to improve readability, we changed the orientation of the page, and we will make sure to request to the publishing office to print the figure in full page and in this same orientation also in the final version of the manuscript.

**Technical corrections:**

Line 177-179: The latter may occur when a strong vegetative growth IS observed early in the season during drought years, especially in energy-limited conditions.

Done.

**References**

FAO. (2021). The impact of disasters and crises on agriculture and food security: 2021. Rome: FAO. doi:10.4060/cb3673en

López-Lozano, R., Duveiller, G., Seguini, L., Meroni, M., García-Condado, S., Hooker, J., … Baruth, B. (2015). Towards regional grain yield forecasting with 1km-resolution EO biophysical products: Strengths and limitations at pan-European level. Agricultural and Forest Meteorology, 206, 12–32. doi:10.1016/j.agrformet.2015.02.021

Monteleone, B., Borzí, I., Bonaccorso, B., and Martina, M. (2022). Developing stage-specific drought vulnerability curves for maizeâ ⁻: The case study of the Po River basin. Agricultural Water Management, 69(May), 107713. doi:10.1016/j.agwat.2022.107713

**Reviewer #2**

The manuscripts highlights the important correlation between the satellite-derived fAPAR product and observed cereal yield in Europe, clearly positioning the study and its results in drought impact monitoring efforts. Therefore, this work fits the journal's scope and can be seen as a timely contribution to the growing early warning and climate services research. In particular the unique scope of the paper, looking at longer, homogeneous periods with robust, significant correlations makes for an interesting, novel article that is written well.

We would like to thank the reviewer for her/his positive stance on our research.

While I enjoyed reading it, some part could benefit for extra clarifications. Besides, I would like to propose a few comments/ideas which could maybe be considered as additional discussion points to strengthen and broaden the manuscript.

- I wonder if the Corine land cover map arable land area is similar to the Eurostat cereal production area data for each NUTS2. It would strengthen the method if this is the case

The cereal production area in Eurostat varies for each year, and a matching between these data and the Corine map goes beyond the scope of the research. The land cover map is mainly used to exclude from the analysis those areas that are clearly not agriculture, such as rural urban areas and forest. We clarified this point in the revised version of the manuscript.

- Could you please better justify why you would take insignificant (but "at least different than zero") values into account (doesn't this reduce precision of the analysis?) and why the particular threshold of 0.15 was chosen?

Indeed, the first analysis was focused only on the statistical significant values (p = 0.05). However, we notice that over some regions statistically significant values were very limited. The inclusion of "at least different than zero" conditions allows for a second-tier of accuracy (with a lower statistical significance) to distinguish between positive and negative correlations. Estimates in this second tier are clearly less accurate, but they provide an additional set of data to interpret the results. This difference in the precision is accounted thought the text, see for instance the matrix adopted in Figure 8 where the relationships over regions with statistically significant correlations are considered more accurate.

In the new version of the manuscript, we further stressed the role of these two complementary information, and why the value 0.15 (roughly 1/3 of the statistically significant value at p = 0.05 for the full sample) was chosen.

- I do not fully understand the following part in the method (L182): "Starting with a minimum length of 2 dekads, up to 990 periods of various length (L, from 2 up to 45 dekads) can be analyzed for each region, and for each of these periods four main metrics are computed". In the rest of the manuscript, it seems the analysis is done on a 1 dekad level (then only combining dekads based on their correlation, not for calculating the correlation).

You are correct, the correlation values are computed on a 1 dekad level. The 990 periods refers to the possible combinations of these 1 level correlation values. In a 45 dek periods, you can combine the r value over 44 possible periods of length 2 deks, 43 periods of length 3, etc.

We reworded the paragraph to clarify the procedure.

- It would strengthen the full analysis to check for sensitivity regarding the chosen -1 thresholds for the drought conditions (L202 and onwards). While not a necessary addition, the paper mainly focusses on its use in drought monitoring systems hence it would be interesting if a similar result would be obtained with other standard deviations as thresholds.

The threshold -1 is a quite common value used in many drought studies, and here it is only used in the final analysis to distinguish between drought and no-drought conditions (now renamed as low yield years and other years). While we agree that the sensitivity of the results to this choice may be interesting, we need to consider the limited length of the available time series (18 years in the regions with no missing values). Hence, a value of -1 select 2 to 3 extreme years from the time series. On the one hand, a larger threshold (in absolute value) may focus only on the most extreme year, reducing the value of the analysis, while on the other hand a smaller (absolute) value may not be representative of extreme conditions.

Given the limited role of this choice in the study, we prefer to do not add any further analysis in this regard, but we better clarify the reasoning behind this choice.

Another sensitivity that could be considered to be evaluated is the detrending method.

We indeed explored also a linear detrending, but we found no significant differences in the results and overall message. We clarified this point in the revised version.

- Besides, in this regard (L205), I wonder why drought years are defined as yield anomaly years (these are agricultural impact years, maybe caused by droughts but potentially by other shocks). The authors could consider reversing the analysis, looking at the average yield anomaly during years with a low average fAPAR during the optimal period. This would not guarantee excluding other shocks (that might impact fAPAR too) but would be more interesting in terms of its capacity to be used as an impact monitoring of prediction product

(also give insight on the FA rate for example). (Similar remark could be made for Figure 2: it shows that indeed, during large drought episodes in Europe, the fAPAR is low, but does not show anything about potentially low fAPAR values during years not considered droughts)

First of all we would like to clarify that Figure 2 reports yield anomalies, not fAPAR anomalies, precisely to verify if yield anomaly patters resemble the expected drought patters. However, we agree with your statement that yield anomalies can be caused by other shocks, not only drought, and we stressed this in the revised version of the manuscript.

As a result, we reworded some parts of the text and reshaped the full section on the yield anomalies to better highlight the message that most (but not all) patterns in yield anomalies corresponds to drought events, so the relevancy of this study for drought monitoring, but that yield anomalies (but also fapar anomalies) cannot be always used as a direct proxy of drought impacts.

Beside this clarification, we agree that the last part of our study can be performed also in reverse, but there are some practical limitations that make this option less straightforward.

By using yield anomalies to discriminate drought/no-drought years (now renamed low/others yield years), we fix the dataset used for each regions independently from the fAPAR period adopted (optimal vs. full years). This means that the same number of low yield years and other years is used for the two comparisons. By starting from fAPAR, the distinction low/others (low fAPAR in this case) will differ for the two aggregation periods, especially in the number of years associated to each category, and this is also further complicated by presence of missing values in the yield tie series. This approach would then make the intercomparison between the results for the two aggregation periods much more difficult to consistently perform.

We now briefly mentioned the consistency as a factor in selecting the approach adopted for this analysis in the revised version of the manuscript.

- I wonder what happened if two periods with Fp+=1 are equally long? Could you please explain how this is handled in the analysis?

This did not occurred very often, since data are at relatively high temporal resolution (10 days) and the temporal behavior of the correlogram is smooth in most of the cases (i.e. bell shaped in many cases, as shown in Figure 1). However, in those rare instances we explored the neighbor NUTS2 and chosen the period closer to the surrounding. This also helped a little to obtain smoother spatial patterns in the outcomes.

We clarified this procedure in the revised version of the manuscript.

- In the method, multiple exclusion criteria (related to fAPAR and EUROSTAT data) are stated, however the results show a full map without data gaps. Does that mean no NUTS2 were excluded based on these criteria?

Even if the Eurostat data used in this study are at NUTS2 level, they are often provide by national authorities. Hence, the availability usually shows national patterns (i.e. rarely a nation provide data only for some NUTS2). In the final outputs, it is possible to see many NUTS2 masked in full countries (e.g. Norway, Switzerland, among other). Of the 267 regions considered at the beginning of the study, only 240 were included in the analysis. We added this statistic to the revised version on the manuscript.

- In figure 4; it is a pity no spatial signal could be visualized. It would be a great addition to show which region of Europe contributes to what here.

As stated in the text, while the length of the homogeneous periods show some clear patters (see Figure 3), the interpretation of the start and end date is much more difficult, mainly due to the large variation in the length of the homogeneous period.

This problem is partially overcome by the next analysis (and in particular by Figure 7) where some spatial patterns in the start and end date of the optimal periods can be deducted.

We better stressed this point in the revised version of the manuscript.

- Technical remark: "It is possible to observed two "flexing points" …" à OBSERVE

Corrected.

- I feel the method behind figure 7 could be explained better. Could the authors please reflect on this choice for the bounded length of 6 to 8 months (rather than only the optimal, correlated period)?

We had to introduce a minimum length, otherwise the optimal average correlation will always be achieved, by definition, by a single dek period corresponding to rmax (or often by 2 deks around the peak if single dekad are excluded). Similarly, an upper boundary was introduce to the optimal length to avoid very long optimal periods over regions with flat signals in the correlogram. We added these considerations in the revised version of the text.

- I wonder if the overall limited correlation in central Europe might be caused by changing crop types over the years / fAPAR is calculated based on the full period thus assuming homogeneity in land cover over this period.

While this is certainly a possible plausible explanation, we think that this can justify some local effects but not the very large areas with limited correlations obtained in this study. Following the suggestion of reviewer #1, we added a further comparison with the results of a similar study at European scale, and we support the conclusion that a possible explanation is the fact that the relationship between fAPAR and yield signals may be less straightforward in those regions where water is not the main limiting factor.

We added these considerations to the revised manuscript.

- I would like for the authors to better explain figure 8: what is meant with performance? Here, how are Fp+/ F+ calculated? Based on the average fAPAR over the optimised period? I think this is missing in the method.

The Fp+/ F+ reported in this figure are the fraction of dekads with statistically significant (or at least positive) r values within each optimal periods. As an example: a Fp+ = 0.7 means that 70% of the dekads in the optimal periods have statistically significant r values. By definition, F+ is always greater than Fp+. We revised the method section to better clarify this calculation.

- In the discussion (eg L363), it seem the authors equate growing season with the season where fAPAR relates to the yield; but can this be supported by agronomic observations? Is this correlation not a sign of a crop growing period vulnerable to droughts rather than a representation of the full season?

Our methodology detect homogenous/optimal periods independently from the growing season but based on the correlation between fAPAR and yield anomalies. The goal is to detect which fAPAR values along the year are better at capturing the final yield in of each year.

The discussion on growing season is a consequence of the obtained results, which highlight that the dekads better capturing yield variations in some cases align well with the growing season. However, this is not always the case and we also clearly stated that "…while studies on satellite-derived phenology have detected growing season lengths ranging from 5 to 9 months… the average length of the periods with positive and statistically significant correlations seems to be shorter" or "Central Europe is characterized by an early start of the optimal period (March to August) that seems to precede the expected growing season (June to October)".

We further stressed in the new version of the manuscript how the optimal period and the growing season do not equate, but they still often overlap in some cases.

- In the discussion (L396-400); the interpretation is not extremely clear. So the inverse relationship between fapar and yield is a result of hot-dry summer months (dry spell months or all months?) and their lagged effect: how? Is this correctly interpreted?)

We reworded the paragraph to clarify this statement. A possible explanation is that there is a lagged response in the fAPAR signal to a drought occurring early in the season, which is reflected in early positive anomalies in fAPAR followed by successive negative anomalies (i.e. a shift in the phenological cycle). This may justify the statistically significant negative correlation early in the season, followed by statistically significant positive correlation later on.

---

## Author Response (AR2)

**Referee #1**

I appreciated the improvements the authors made to the manuscript following the suggestions of both reviewers. The authors exhaustively answered all the questions I raised in my previous report.

Below just minor comments and technical corrections.

Lines 351-355: is there an explanation for the fact that Spain exhibits a robust performance, while mixed results can be observed for Italy and Greece? Is this related with the quality of yield data?

R: A possible explanation of these mixed results can be the complex morphology of some regions in Italy and Greece, where the remote sensing data may perform poorly. We detailed this observation in:

L462-465 "Given the complex morphology of those regions, potential unreliability in the fAPAR estimates may be a possible cause for the poor performances. Complex morphology can also be the reason for poor results over few other Mediterranean areas, such as Greece."

Of course, we cannot exclude differences in data quality within Eurostat, but we do not have enough information for such assessment.

Section 3 is "Results and Discussion" while Section 4 is "Discussion". I suppose Section 3 should be "Results" and Section 4 "Discussion"

R: Yes, thanks for detecting the error.

Line 360: I suppose "where" should be changed with "were"

R: Yes, corrected.

Line 384: vales should be "values"

R: Yes, corrected.

Page 28 is empty.

R: The empty page has been removed.